# Motility-induced coexistence of a hot liquid and a cold gas

Lukas Hecht [1], Iris Dong [1] & Benno Liebchen [1]✉

If two phases exist at the same time, such as a gas and a liquid, they have the same temperature. This fundamental law of equilibrium physics is known to apply even to many non-equilibrium systems. However, recently, there has been much attention in the finding that inertial self-propelled particles like Janus colloids in a plasma or microflyers could self-organize into a hot gas-like phase that coexists with a colder liquid-like phase. Here, we show that a kinetic temperature difference across coexisting phases can occur even in equilibrium systems when adding generic (overdamped) self-propelled particles. In particular, we consider mixtures of overdamped active and inertial passive Brownian particles and show that when they phase separate into a dense and a dilute phase, both phases have different kinetic temperatures. Surprisingly, we find that the dense phase (liquid) cannot only be colder but also hotter than the dilute phase (gas). This effect hinges on correlated motions where active particles collectively push and heat up passive ones primarily within the dense phase. Our results answer the fundamental question if a non-equilibrium gas can be colder than a coexisting liquid and create a route to equip matter with self-organized domains of different kinetic temperatures.

We are all used to the experience that a gas is often hotter than a liquid of the same material. For example, to evaporate water from a pot in our kitchens, we need to increase its temperature. Then, at some point, vapor molecules rapidly escape from the liquid and distribute in the surrounding air. This experience that vapor emerges when increasing the temperature of a liquid has played a key role throughout human history: It was an essential ingredient, e.g., for the development of the steam engine[1], and it is key to technological applications like distillation techniques[2,3] or physical vapor deposition[4,5] as well as to natural spectacles such as geysers[6]. The central exception from the experience that gases are hotter than liquids of the same material occurs when two phases, e.g., a gas and a liquid, coexist at the same time. Then they share the same temperature. This is guaranteed by the fundamental laws of statistical mechanics and thermodynamics for all equilibrium systems and it also applies to some non-equilibrium systems[7–11]. Intuitively, this is plausible since any type of temperature gradient seems to evoke an energy flow evening out an initial temperature gradient.

Despite this, very recently, it was found that at phase coexistence in certain active systems consisting of active particles which consume energy from their environment to propel themselves[12–14], the dilute (gas-like) phase is hotter by up to one or two orders of magnitude compared to the dense (liquid-like) phase in terms of the kinetic temperature[11,15,16]. (Here, the kinetic temperature is defined as the mean kinetic energy per particle, which is equivalent to other temperature definitions in equilibrium.) While this complies with our intuition that gases are often hotter than liquids, it is in stark contrast to the situation in equilibrium systems and the expectation that any temperature difference should evoke an energy flux that balances it out. By now, such a temperature difference across coexisting phases has been shown to occur for a variety of temperature definitions that all coincide in equilibrium. It has been observed, e.g., for the kinetic temperature[11], the effective temperature[15] as well as for tracer-based temperature definitions[16,17] in systems undergoing motility-induced phase separation (MIPS)[7–9,18–30], i.e., in systems of active particles that self-organize into a dilute (gas) and a coexisting dense (liquid) phase.

[1]Institute of Condensed Matter Physics, Department of Physics, Technical University of Darmstadt, Darmstadt, Germany.
✉e-mail: benno.liebchen@pkm.tu-darmstadt.de

The mechanism underlying the emergence of a temperature difference across coexisting phases hinges on the consumption of energy at the level of the active particles when undergoing frequent collisions within the dense phase. This mechanism crucially requires inertia[11,15,16], whereas overdamped active particles show the same kinetic temperature in coexisting phases[11]. The requirement of inertia restricts the observation of different coexisting temperatures to a special class of active systems and precludes its experimental observation in generic microswimmer experiments.

In this work, we explore the possibility to achieve a kinetic temperature difference across coexisting phases in mixtures of two components that on their own would not lead to a temperature difference: an ordinary equilibrium system made of inertial passive Brownian tracer particles such as granular particles or colloidal plasmas and overdamped active Brownian particles like bacteria or synthetic microswimmers. Our exploration leads to the following central insights: first, we show that when the mixture undergoes MIPS, the passive particles in the dense and the dilute phase indeed can have different kinetic temperatures (and different Maxwell–Boltzmann temperatures, which are defined based on the width of the velocity distribution). This demonstrates that kinetic temperature differences in coexisting phases can occur in a broader class of systems than what was anticipated so far. Second, we find that not only the gas can be hotter—but, counterintuitively—also the dense phase can be hotter than the dilute phase. This appears particularly surprising since the current understanding of the mechanism leading to different temperatures across coexisting phases in pure active systems hinges on the idea that frequent directional changes due to collisions lead to a local loss of kinetic energy (similarly as inelastic collisions do in granular systems[31–43]). Such collisions are more frequent in dense regions suggesting that the dense phase is always colder than the dilute one, which coincides with all previous observations[11,15,16,31–42,44]. We find that this mechanism also applies to the passive tracers of the active–passive mixtures studied here in a certain parameter regime in which the tracers are trapped within dense clusters by surrounding active particles leading to low tracer temperatures in dense regions analogously as in the single-species case of inertial active particles. However, surprisingly, we find that this effect can also be reverted in mixtures of active and passive particles. This is because for strong self-propulsion, the active particles persistently push passive ones forward even within the dense phase, which can overcompensate the slowing of the latter ones due to (isotropic) collisions with other particles. Hence, the passive particles can achieve a higher (kinetic) temperature in the dense phase than in the surrounding gas, where correlated active–passive motions occur less frequently and last shorter. Our results pave the route towards the usage of microswimmers such as bacteria[45–49], algae[50,51], or Janus particles and other synthetic microswimmers[12,20,52–54] for controlling the kinetic temperature profile and hence, the dynamics of fluids and other passive materials.

## Results

### Model

We study a mixture of active and passive particles in two spatial dimensions, in which the active (passive) particles are represented by the active (passive) Brownian particle [ABP (PBP)] model[11,26,30,55–57]. While the motion of the active particles is overdamped, the passive species is significantly heavier (inertial). For simplicity, we consider active and passive particles with the same size and drag coefficients[58] but different material density. However, note that the key effects which we discuss in the following are similar for particles with significantly different sizes and drag coefficients, as we shall see. The particles are represented by (slightly soft) spheres and the dynamics of the active particles is made overdamped by choosing a very small mass $m_a$ and a small moment of inertia $I = m_a \sigma_a^2 / 10$ (corresponding to a rigid sphere). The active particles feature an effective self-propulsion force

$\mathbf{F}_{SP,i} = \gamma_t v_0 \mathbf{p}_i(t)$, where $v_0$, $\mathbf{p}_i$ denote the (terminal) self-propulsion speed and the orientation $\mathbf{p}_i(t) = (\cos \phi_i(t), \sin \phi_i(t))$ of the $i$th active particle ($i = 1, 2, \ldots, N_a$), respectively. The position $\mathbf{r}_i$ and the orientation angle $\phi_i$ of the $i$th active particle evolve according to $d\mathbf{r}_i/dt = \mathbf{v}_i$ and $d\phi_i/dt = \omega_i$, respectively, where the velocity $\mathbf{v}_i$ and the angular velocity $\omega_i$ evolve as

$$m_a \frac{d\mathbf{v}_i}{dt} = -\gamma_t \mathbf{v}_i + \gamma_t v_0 \mathbf{p}_i - \sum_{\substack{n=1 \\ n \neq i}}^{N} \boldsymbol{\nabla}_{\mathbf{r}_i} u(r_{ni}) + \sqrt{2 k_B T_b \gamma_t} \boldsymbol{\xi}_i, \tag{1}$$

$$I \frac{d\omega_i}{dt} = -\gamma_r \omega_i + \sqrt{2 k_B T_b \gamma_r} \eta_i. \tag{2}$$

Here, $T_b$ represents the bath temperature, $\gamma_t$ and $\gamma_r$ are the translational and rotational drag coefficients, respectively, and $k_B$ is the Boltzmann constant. To have access to a well-defined instantaneous particle velocity, we explicitly account for inertia for the active species but choose a very small mass to stay in the overdamped regime. Notice that using overdamped Langevin equations instead ($m_a = 0$) essentially yields the same results (Fig. S13, Supplementary Information).

The passive particles feature a comparatively large mass $m_p$ and their velocity $\mathbf{v}_j$ evolves as

$$m_p \frac{d\mathbf{v}_j}{dt} = -\tilde{\gamma}_t \mathbf{v}_j - \sum_{\substack{n=1 \\ n \neq j}}^{N} \boldsymbol{\nabla}_{\mathbf{r}_j} u(r_{nj}) + \sqrt{2 k_B T_b \tilde{\gamma}_t} \boldsymbol{\xi}_j \tag{3}$$

with particle index $j = N_a + 1, N_a + 2, \ldots, N_a + N_p$ and drag coefficient $\tilde{\gamma}_t$. The interaction potential $u(r_{nl})$, $r_{nl} = |\mathbf{r}_n - \mathbf{r}_l|$ is modeled by the Weeks–Chandler–Anderson (WCA) potential[59], and $\boldsymbol{\xi}_{i/j}$ and $\eta_i$ denote Gaussian white noise with zero mean and unit variance. Equations (1)–(3) are solved numerically by using LAMMPS[60] (see "Methods" for details). Finally, we define the Péclet number, which measures the relative importance of self-propulsion and diffusion, by $Pe = v_0 / \sqrt{2 D_r D_t}$, where $D_t = k_B T_b / \gamma_t$ and $D_r = k_B T_b / \gamma_r$ denote the translational and the rotational diffusion coefficients of the active particles, respectively.

### Coexistence of a hot gas and a cold liquid

Let us first consider an initially uniform distribution of an overdamped mixture of active and passive particles[58,61,62]. In our simulations with $Pe = 100$, an area fraction of $\varphi_{tot} = 0.5$, and a fraction of $x_a = 0.6$ active particles, we observe that the active and passive particles aggregate and form persistent clusters despite the fact that they interact purely repulsively. These clusters are motility-induced[7–9,18–26,29,30] and continuously grow (coarsen), ultimately leading to a phase-separated state comprising a dense liquid-like region that coexists with a dilute gas phase (Fig. 1a–d and Movie S1, Supplementary Information), which is in agreement with previous studies[58,61]. As for systems of active overdamped particles alone[11], we find that the active and passive particles in both phases have the same kinetic temperature (shown in Fig. 1d for the passive particles). Here, following refs. 11,16,63,64, we define the temperature of the particles based on their kinetic energy as $k_B T_{kin}^{a/p} = m_{a/p} \langle (\mathbf{v} - \langle \mathbf{v} \rangle)^2 \rangle / 2$, which is well-defined also in non-equilibrium systems[65]. (Note that the phenomena which we report occur similarly if using other temperature definitions such as the Maxwell–Boltzmann temperature, as further discussed below.) Let us now explore if the situation found for the overdamped mixture changes when replacing the overdamped tracers with (heavier) underdamped ones (Fig. 1e–h). Then, at the level of the structures that emerge, not much changes in our simulations: we still observe the formation of small clusters, which is followed by coarsening, ultimately

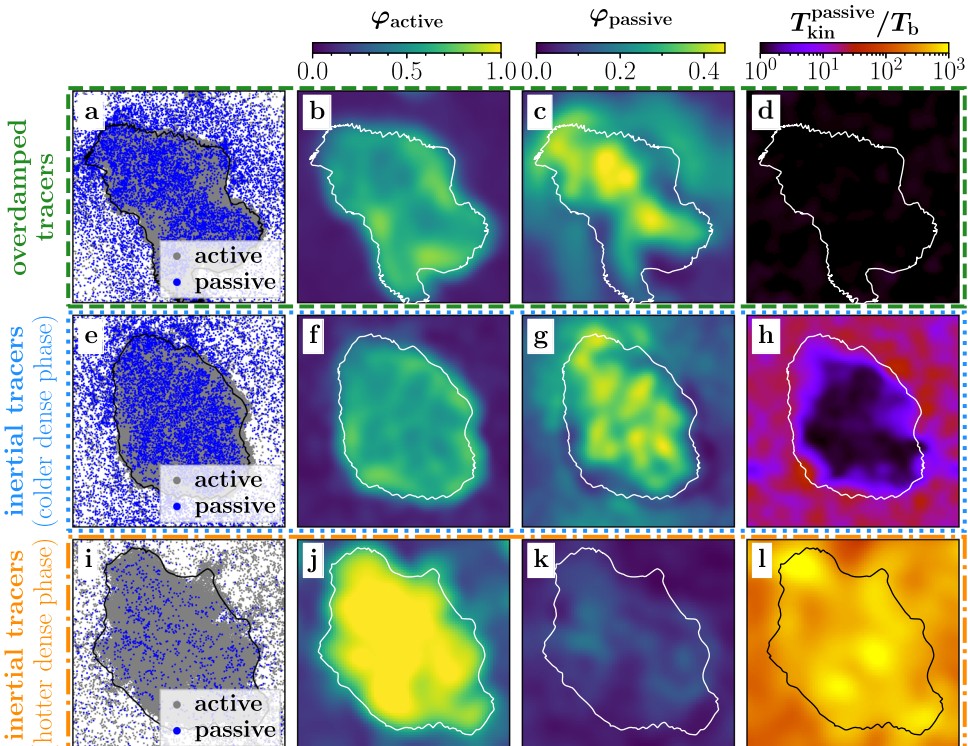

**Fig. 1 | Kinetic temperature and area fraction at coexistence.** From left to right we show a snapshot of the particle positions in the steady state (**a, e, i**), the local area fraction of active (**b, f, j**) and passive (**c, g, k**) particles, and the coarse-grained kinetic temperature field of the passive tracer particles (**d, h, l**) averaged over time in the steady state, respectively. **i**–**l** are slightly zoomed in and the black and white solid lines are guides to the eye denoting the border of the dense phase. Parameters: $x_a = 0.6$, Pe $= 100$, $m_p/(\bar{\gamma}_t \tau_p) = 5 \times 10^{-5}$ (**a**–**d**); $x_a = 0.6$, Pe $= 100$, $m_p/(\bar{\gamma}_t \tau_p) = 5 \times 10^{-2}$ (**e**–**h**); $x_a = 0.9$, Pe $= 400$, $m_p/(\bar{\gamma}_t \tau_p) = 5 \times 10^{-2}$ (**i**–**l**); $N_a + N_p = 20\,000$, $\varphi_{tot} = 0.5$.

leading to complete phase separation. However, when exploring the kinetic temperature of the passive particles within the steady state, we find that, remarkably, the passive particles in the dense phase are colder than in the dilute phase. The temperature ratio of the two phases is highly significant and amounts to ~2.5 (Fig. 1h and Movie S2, Supplementary Information). While this temperature difference is similar to what has previously been seen in underdamped active particles[11,15,16] and driven granular particles[31–42,44], its emergence in the present setup is surprising since it is well known that neither the overdamped active particles[11] nor the underdamped tracers alone[66,67] would result in a kinetic temperature difference across coexisting phases. Accordingly, the temperature difference must arise from the interactions of the two species. To understand this in detail, it first might be tempting to start from the common understanding of kinetic temperature differences in granular systems or purely active systems made of inertial ABPs such as Janus colloids in a plasma[68], microflyers[69], or beetles at interfaces[70] which relates the emergence of a temperature difference to an enhanced energy dissipation in the dense phase at the level of the particles. The latter could occur due to inelastic collisions as for granular particles[39] or due to multiple collisions between which drag forces transfer energy from the particles to the surrounding liquid as for active particles[11,16]. However, in the present case of a mixture, collisions between active and passive particles have a different effect in the dense and in the dilute phase. In the dense phase, the motion of the passive particles is constricted by the surrounding clustered ABPs (see, e.g., Fig. 1e), which accumulate mostly at the border of the clusters, similarly as in completely overdamped mixtures[58,61], and which cause an effective attraction between the passive tracers by pushing them together[49,71,72] (see also supplementary text and Fig. S12, Supplementary Information). Therefore, the passive particles cannot move much in the dense phase and have a

lower kinetic energy there compared to the dilute phase. This is also visible in the velocity distribution of the passive particles, which narrows for increasing $x_a$ in the dense phase (Fig. S1, Supplementary Information). In contrast, in the dilute phase, when active particles collide with passive particles, they can persistently push passive particles forward and accelerate them such that their kinetic energy increases above the energy they would have due to the surrounding heat bath. Such a correlated active–passive dynamics heats up the passive particles in the dilute phase and leads to a broader velocity distribution at intermediate Pe, such as Pe $= 100$ (see below).

## Hot liquid-like droplets in a cold gas

Since the observed temperature differences are activity-induced, one might expect that the temperature gradient further increases when enhancing the self-propulsion speed of the active particles, i.e., when increasing Pe. Surprisingly, however, in many cases, the opposite is true. For example, for fractions $x_a = 0.3$, 0.6, or 0.9 of ABPs, we find that the kinetic temperature difference is largest for some intermediate Pe and then decreases essentially monotonously with increasing Pe (Fig. 2) before it even reverts, and we obtain dense liquid-like droplets that are hotter than the surrounding gas. As time evolves, these droplets grow (coarsening) leading to larger and larger clusters, ultimately resulting in a single hot and dense cluster that persists over time. Exemplarily, we show typical snapshots for the case Pe $= 400$, $x_a = 0.9$ in Fig. 1i–l (see also Movie S3 and Fig. S2, Supplementary Information). In panel l, one can clearly see that the liquid in the center of the figure is hotter than the surrounding gas. Such a coexistence of a hot liquid-like droplet and a cold gas—in terms of the kinetic temperature—is in stark contrast to what has been found for underdamped active particles[11,15,16] and for driven granular particles[31,32,39–42]. Note that for very large liquid-like droplets containing significantly more than

about $10^4$ particles, it may happen that not the entire droplets are hot but only a certain layer at their boundaries. The emergence of a hot dense droplet also contrasts with the intuitive picture given above that hinges on the idea that active particles can efficiently push forward and accelerate passive particles only in low-density regions. Therefore, the key question that guides our explorations in the following is: What is the mechanism allowing for a coexistence of hot liquid-like droplets and a colder gas?

## Mechanism: correlated active–passive dynamics heats tracers in the dense phase

We now explore the mechanism underlying our previous observations that in mixtures of overdamped ABPs and inertial passive particles dense liquid-like droplets are persistently hotter than the surrounding gas at large Pe. To this end, we first analyze the velocity distribution of the passive tracers in the uniform regime at $x_a = 0.2$ and in the phase-separated regime at $x_a = 0.8$, which broadens as Pe increases (Fig. 3a–c). Strikingly, if and only if the active particles are sufficiently fast (Pe $\gtrsim$ 200), the velocity distribution broadens more in the dense phase than in the dilute phase (Fig. 3b–d). This means that increasing the speed of the active particles (i.e., increasing Pe) has a much stronger effect on the speed of the passive particles in the dense regime (where collisions are more frequent) than in the dilute regime, which ultimately leads to hot liquid-like droplets. What remains open at this stage is why the velocity distribution broadens faster for passive particles in the dense regime than in the dilute regime (only) if the Péclet number is large.

To answer this question, we now explore the power balance of the passive particles in the dense and the dilute phase. As we will see, this power balance will point us to correlations between active and passive particles which lead to hot liquid-like droplets at large Pe. To obtain a

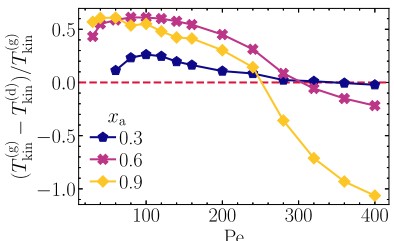

**Fig. 2 | Kinetic temperature difference.** Normalized kinetic temperature difference $(T_{kin}^{(g)} - T_{kin}^{(d)})/T_{kin}^{(g)}$ with mean kinetic temperatures of passive tracers $T_{kin}^{(g)}$ and $T_{kin}^{(d)}$ in the gas and the dense phase, respectively, as a function of Pe for three different values of $x_a$ as denoted in the key. Parameters: $m_a/(\gamma_t \tau_p) = 5 \times 10^{-5}$, $m_p/(\bar{\gamma}_t \tau_p) = 5 \times 10^{-2}$, $\varphi_{tot} = 0.5$, $N_a + N_p = 20\,000$.

power balance equation for the passive particles, we first multiply Eq. (3) by $\mathbf{v}_j$ and take the ensemble average. With $k_B T_{kin} = m_p \langle \mathbf{v}^2 \rangle / 2$, this leads to

$$\frac{2\bar{\gamma}_t}{m_p} k_B T_{kin} = \frac{2\bar{\gamma}_t}{m_p} k_B T_b + \langle \mathbf{v} \cdot \mathbf{F}_{int} \rangle \tag{4}$$

in the steady state, where $\mathbf{F}_{int,j} = -\sum_{\substack{n=1 \\ n \neq j}}^{N} \nabla_{\mathbf{r}_j} u(r_{nj})$ is the total interaction force on particle $j$ and $r_{nj} = |\mathbf{r}_n - \mathbf{r}_j|$. If we now compare the power balance for particles in the dense and in the gas phase, we can express the kinetic temperature difference as

$$k_B \left( T_{kin}^{gas} - T_{kin}^{dense} \right) = \frac{m_p}{2\bar{\gamma}_t} \left[ \langle \mathbf{v} \cdot \mathbf{F}_{int} \rangle_{gas} - \langle \mathbf{v} \cdot \mathbf{F}_{int} \rangle_{dense} \right]. \tag{5}$$

This central equation leads to two important conclusions: first, the kinetic temperature difference between the dense and the gas phase is proportional to $m_p/\bar{\gamma}_t$, which vanishes if the passive particles are overdamped in accordance with our simulations (Fig. 1d and S3, Supplementary Information). Interestingly, the same proportionality has also been observed for a single-component system consisting of inertial ABPs, where it has been observed that the dense phase is always colder than the gas phase[11]. Second, the temperature difference depends on the interaction between the particles given by the term $\langle \mathbf{v} \cdot \mathbf{F}_{int} \rangle$, which measures how strongly interactions push passive particles forward in their direction of motion. From the probability distribution of the individual values $\mathbf{v} \cdot \mathbf{F}_{int}$ that contribute to the mean (Fig. 4a, e), we obtain significant differences between the dense and the gas phase at large values which determine the sign of the temperature difference: at intermediate Pe, e.g., Pe = 80 (Fig. 4a), large values of $\mathbf{v} \cdot \mathbf{F}_{int}$ are more frequent in the gas phase than in the dense phase (see also Fig. 4d). That is, events in which the interaction force and the velocity of the passive particles are aligned and large (e.g., if an ABP is pushing a passive particle forward[73]) are more frequent in the gas phase than in the dense phase, in which the particles have significantly less space to move and accelerate. In contrast, at large Pe, such events are more frequent in the dense phase finally leading to the coexistence of hot liquid-like droplets with a colder gas (Fig. 4e, h). Intuitively, this is because at very large Pe, ABPs can (collectively) push passive particles forward over relatively long periods of time even in the dense phase without being stopped by collisions with other particles due to the strong effective self-propulsion force (cf. Movie S4, Supplementary Information). These correlated particle dynamics are exemplarily shown in Fig. 4b, f and schematically visualized in Fig. 5. The correlated dynamics of active and passive particles also lead to a long ballistic regime in the mean-square displacement of the passive particles at

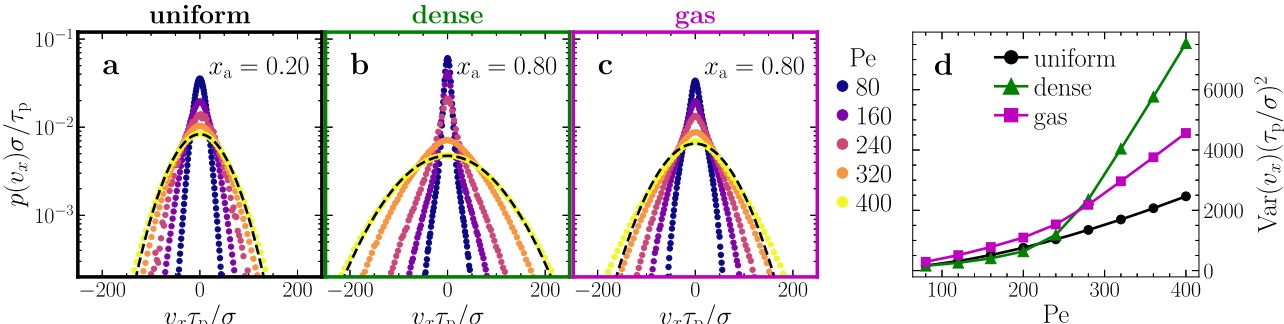

**Fig. 3 | Velocity distribution of passive tracers.** Distribution of the $x$ component $v_x$ of the passive particles' velocities **a** in the uniform state at $x_a = 0.2$ and **b**, **c** in the phase-separated state at $x_a = 0.8$ for passive particles in the dense and the gas phase, respectively, averaged over time in the steady state. Different Pe values are given in the key (parameters as in Fig. 2). The black dashed lines are Gaussian fits. **d** Variance of $v_x$ as function of Pe for the three cases shown in (**a–c**) averaged over five independent ensembles in the steady state.

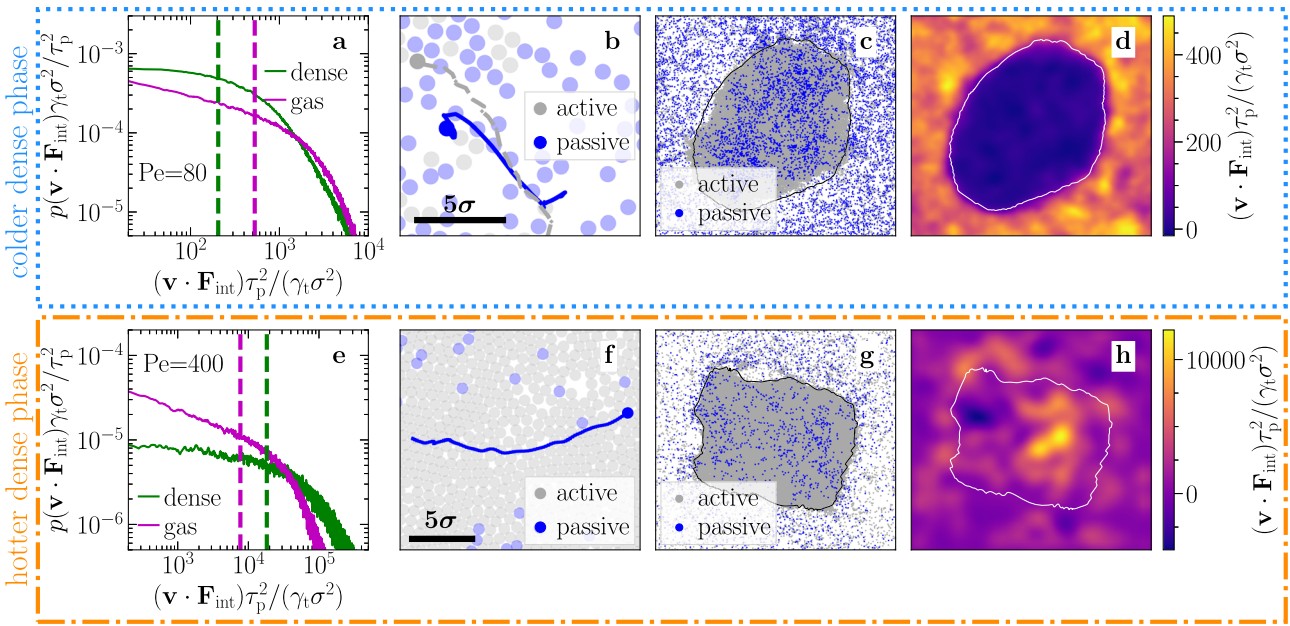

**Fig. 4 | Correlated particle dynamics. a, e** Distribution of $\mathbf{v} \cdot \mathbf{F}_{\text{int}}$ for the passive tracers (PBPs) averaged over time in the steady state. The dashed vertical lines mark the corresponding mean values. **b** Exemplary trajectories of an active and a passive particle in the dilute phase (gray dashed and blue solid line, respectively), where the active particle pushes the passive particle forward such that $\mathbf{v}$ and $\mathbf{F}_{\text{int}}$ are aligned and $\mathbf{v} \cdot \mathbf{F}_{\text{int}}$ is large. **f** Exemplary trajectory of a passive particle in a liquid-like droplet pushed forward as a result of correlated dynamics of the active particles. **c, g** Snapshots of the corresponding simulations in the steady state. **d, h** Corresponding coarse-grained values of $\mathbf{v} \cdot \mathbf{F}_{\text{int}}$. The black and white solid lines in panels **c**, **d**, **g**, and **h** indicate the border of the dense phase. Parameters: Pe = 80 and $x_a = 0.7$ (**a, c, d**), Pe = 100 and $x_a = 0.6$ (**b**), Pe = 400 and $x_a = 0.9$ (**e–h**); other parameters as in Fig. 2.

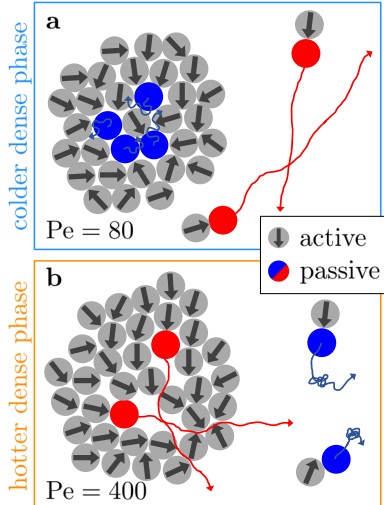

**Fig. 5 | Schematic illustration of the mechanism. a** At low Pe, particles are trapped in the dense phase and have no space to speed up. Long correlated trajectories where active particles push passive ones forward occur only in the dilute phase. Thus, passive particles are faster in the dilute phase. **b** At large Pe (fast self-propulsion), long correlated trajectories where active particles push passive ones forward occur even in the dense phase (and are supported by collective motion of the active particles). The more frequent collisions in combination with the collective motion of the ABPs in the dense phase lead to faster passive particles in liquid-like droplets compared to the surrounding gas.

intermediate times (similar as for a completely overdamped mixture[61]) before the dynamics of the passive particles becomes diffusive again (Fig. S4f, Supplementary Information). Finally, we can ask why the temperature difference between the hot liquid-like droplets and the cold gas is larger at large $x_a$. This is because at large $x_a$, the active particles accumulate in the dense phase and induce stronger collective

motions in that place when they are many (see also Fig. S5, Supplementary Information). Conversely, the fraction of active particles in the surrounding gas does not depend much on $x_a$, and hence, the collision rate in the gas does not increase with $x_a$.

### Non-equilibrium state diagram

Having seen that the coexistence of hot liquid-like droplets and a cold gas requires sufficiently fast self-propulsion of the active particles, i.e., large Pe, we now examine the parameter dependence more systematically. Therefore, we explore the non-equilibrium state diagram by varying Pe ∈ [0, 400] and $x_a$ ∈ [0.0, 1.0] at a constant area fraction $\varphi_{\text{tot}} = 0.5$. The transition line between the uniform and the MIPS regime is obtained by analyzing the local area fraction $\varphi_{\text{loc}}(\mathbf{r}_i) = \sum_{j=1}^{N} \sigma_j^2 H(R - |\mathbf{r}_i - \mathbf{r}_j|)/(4R^2)$, where H is the Heaviside step function and $\sigma_j$ the diameter of particle $j$, calculated from averages over circles of radius $R = 5\sigma$. Its distribution is unimodal in the uniform regime and bimodal in the coexistence regime allowing to distinguish between the uniform and the MIPS regime[25,30,74,75] (Fig. S6, Supplementary Information). We distinguish between the passive particles in the dense and the dilute phase in the steady state and calculate their mean kinetic temperature (see "Methods" and Fig. S7, Supplementary Information, for details). The system phase separates for large enough fraction of active particles $x_a$ and large enough Pe (Fig. 6). At small Pe, the transition line approximately follows the transition line of a purely overdamped mixture as obtained in ref. 58, which reads $x_a^{(\text{critical})} \propto 1/(\varphi_{\text{tot}} Pe)$. However, at large Pe, the partially underdamped system requires a larger fraction of active particles to undergo MIPS than the purely overdamped system, which can be understood as a consequence of inertial effects: at large Pe, passive particles are typically fast when they collide with an ABP. Due to their inertia, the passive particles slow down only gradually and sometimes even push aggregated ABPs apart, which can destroy small aggregations. This effect is particularly pronounced for large Pe and opposes the onset of MIPS. Hence, compared to a completely overdamped system, a larger fraction of active particles is required to initiate MIPS at large Pe.

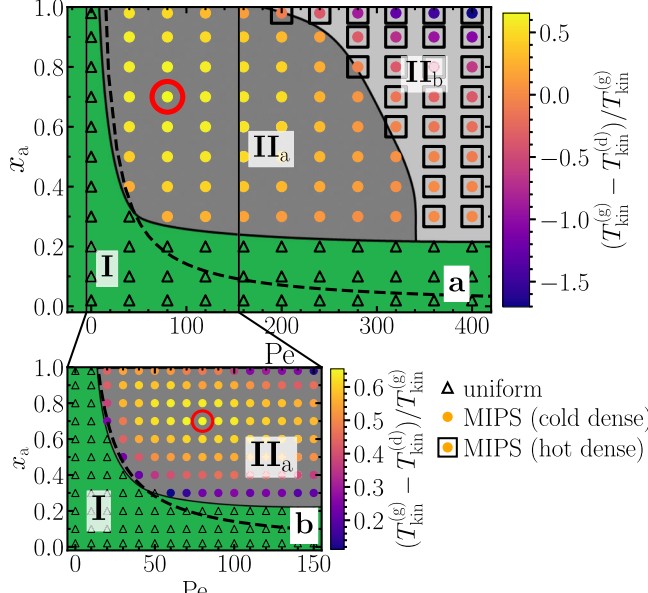

**Fig. 6 | Non-equilibrium state diagram for $2 \times 10^4$ particles. a** (I) uniform state, (II$_a$) hot gas and cold liquid coexistence, and (II$_b$) cold gas and hot liquid-like droplets coexistence. The colors denote the (normalized) kinetic temperature difference of the passive particles in the dense and the gas phase. The black dashed line denotes the transition line as obtained in ref. 58 for a purely overdamped mixture. **b** shows a zoomed version of (**a**) and the red circle denotes the maximum temperature difference in regime II$_a$. All data are averaged over a time interval $\Delta t = 900\tau_p$ in the steady state (parameters as in Fig. 2).

The different kinetic temperatures in the dense and the dilute phase are indicated by the colors in Fig. 6. It can be seen that the temperature difference between the dense and the dilute phase strongly depends on both $x_a$ and Pe: in accordance to the mechanism which we have discussed in the previous section, we find that for intermediate Pe, the dense phase shows a lower kinetic temperature than the dilute phase with a maximum temperature difference around (Pe, $x_a) \approx (80, 0.7)$ (red circle in Fig. 6). For large Pe and large $x_a$, the kinetic temperature difference changes its sign, indicated by the squares in Fig. 6a, i.e., hot liquid-like droplets coexist with a cold gas. The latter occurs at lower Pe for increasing $x_a$ because the overall energy transfer from the active to the passive particles is larger for large $x_a$ only in the dense phase, where the active particles increasingly accumulate as $x_a$ increases (see also Figs. S4 and S5, Supplementary Information). This can also be seen from the parameter dependence of the kinetic temperature of the passive particles (Fig. S8, Supplementary Information): the kinetic temperature increases with increasing $x_a$ (and increasing Pe) in the dense phase but shows a maximum at intermediate $x_a$ in the gas phase, where the fraction of active particles hardly increases when increasing $x_a$ beyond a certain point.

### Role of inertia
Inertia of the passive particles is a key ingredient to observe coexisting temperatures. This can be seen in Fig. S3 (Supplementary Information) and from Eq. (5): the temperature difference is proportional to the ratio $m_p/\tilde{\gamma}_t$. Thus, in the overdamped limit $m_p/\tilde{\gamma}_t \to 0$, the temperature difference vanishes (Fig. 1d) because the passive particles react instantaneously to acting forces. Thus, their motion, and hence, also their kinetic temperature, is dominated by diffusion[11]. In contrast, sufficiently heavy (inertial) tracer particles can store the energy gained during collisions with active particles as kinetic energy such that their kinetic temperature is not determined by diffusion alone, which is fully consistent with our simulation data and previous

literature[11,15,16,31,32,39–42]. Increasing inertia does also lead to a significant violation of the equipartition theorem both in the dense and the gas phase (Fig. S9, Supplementary Information), which indicates that the system is increasingly far away from equilibrium when increasing inertia of the passive particles.

### Role of the particle size
For simplicity, we have considered active and passive particles with the same size and the same drag coefficients so far but with significantly different material density. Now, we show that persistent temperature differences also occur when the passive particles are significantly larger than the active ones. We have varied the size ratio $s = \sigma_p/\sigma_a \in \{1, 2, 3, 4, 6, 8, 10\}$ keeping $\sigma_a$ as well as $m_p$, $m_a$, and $\varphi_{tot}$ fixed. The fraction $x_a$ is chosen such that the area fraction of active particles is approximately 0.5 for small size ratios. For large size ratios, we kept $x_a = 0.99$ fixed to ensure that enough passive particles are inside the system. For the drag coefficient of the passive particles, we choose $\tilde{\gamma}_t = \sigma_p\gamma_t/\sigma_a$. Our results are exemplarily shown in Fig. 7 for Pe = 100. Here, we observe a persistent kinetic temperature difference between the passive particles in the dense and the gas phase even for significantly different particle sizes (Fig. 7h). This temperature difference is also visible in the velocity distributions, which are broader in the gas phase compared to the dense phase (Fig. 7e, f). Hence, the observation of a cold dense phase that coexists with a hotter surrounding gas persists even for large size ratios. The opposite case, i.e., hot liquid-like droplets coexisting with a colder gas, is also robust and leads to a temperature difference of $T_{kin}^{(dense)} - T_{kin}^{(gas)} \approx 196.0\, T_b$ for $\sigma_p/\sigma_a = 10$ at Pe = 400, $x_a = 0.99$, and $\varphi_{tot} = 0.70$ for example. Notice however, that very large passive particles tend to accumulate in the dilute phase especially at large Pe making it challenging to calculate a precise value of the temperature difference. Interestingly, this is in contrast to purely overdamped mixtures, where large size ratios support the formation of large passive-particle clusters[49,72].

### How representative is the kinetic temperature?
So far, following refs. 11,16,63,64, we have used the kinetic energy of the particles to define a kinetic temperature as a measure for the temperature. The kinetic temperature has frequently been used for granular systems[31,32,39–42,76–78] and is also well-defined in non-equilibrium systems[65]. In equilibrium, the kinetic temperature is equal to the thermodynamic temperature[66]. In the binary mixtures of active and passive particles studied in the present work, the kinetic temperature of the passive tracer particles, which measures the velocity fluctuations, has two contributions: one from the thermal Brownian motion and one originating from collisions with surrounding active and passive particles. From the previously discussed results, we know that the latter cause the kinetic temperature difference between passive particles in the dense and the gas phase. In addition, we analyzed the velocity distribution of the passive particles in the dense and the gas phase. The variance of this distribution (Fig. 3d) exhibits the same behavior as the kinetic temperature. Remarkably, the velocity distributions are approximately Gaussian for sufficiently large Pe (Fig. 3a–c). We exploit this to define a Maxwell–Boltzmann temperature $T_{MB}$ by fitting a Maxwell–Boltzmann distribution to the velocity distribution with one free fit parameter $k_B T_{MB}$. For the data shown in Fig. 3b, c at Pe = 400 we obtain $T_{MB}/T_{bath} = 3.6 \times 10^2$ (liquid-like droplets) and $T_{MB}/T_{bath} = 1.9 \times 10^2$ (gas). This shows that mixtures of inertial tracers and overdamped ABPs can lead to self-organized hot liquid-like droplets that coexist with a colder gas also in terms of the Maxwell–Boltzmann temperature.

Since both the kinetic temperature and the Maxwell–Boltzmann temperature are sensitive to local collective motion patterns of the particles and a sensible measure for the temperature of the particles should measure their independent motion, we now explore if spatial

velocity correlations of the passive tracer particles are crucial for the emergence of a temperature difference. For that we calculate the spatial velocity correlation function[79]

$$C_v(r) = \frac{\langle \mathbf{v}(r) \cdot \mathbf{v}(0) \rangle}{\langle \mathbf{v}(0)^2 \rangle}. \tag{6}$$

As shown in Fig. 8b (and Movie S4, Supplementary Information), velocity correlations are indeed present between the passive particles

in the dense phase over a significant spatial range indicating that collective motion might strongly influence the kinetic temperatures. In fact, we find that the mean distance between the passive particles in the dense phase calculated from a Voronoi tessellation is given by ~4.6$\sigma$ for the case shown in Fig. 8a–c and therefore, much smaller than the length scale of the velocity correlations (Fig. 8b). To exclude that such collective motions are required to achieve a coexistence of a hot liquid and a cold gas, we have performed simulations with a very low fraction of passive particles such that their typical distances to each other are significantly longer than the velocity correlations

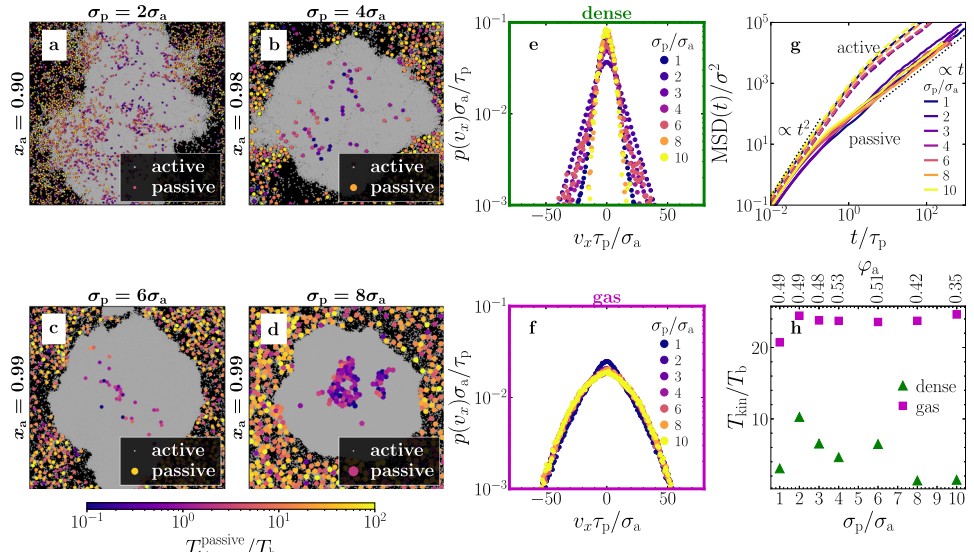

**Fig. 7 | Effect of the particle size. a–d** Simulation snapshots with four different sizes of the passive particles as given in the panel titles. The passive particles are colored with respect to their kinetic temperature. **e, f** Velocity distribution of the passive particles in the dense and the gas phase, respectively, for different size ratios as indicated in the key. **g** Mean-square displacement of the active (dashed lines) and passive (solid lines) particles. **h** Corresponding kinetic temperatures of passive particles in the dense and the gas phase. Parameters: Pe = 100, $\bar{\gamma}_t = \sigma_p \gamma_t / \sigma_a$, $\varphi_{tot} = 0.7$, $N_a + N_p = 2 \times 10^4$ for $\sigma_p/\sigma_a = 1, 2, 3, 4$ and $N_a + N_p = 5 \times 10^4$ for $\sigma_p/\sigma_a = 6, 8, 10$ (other parameters as in Fig. 2).

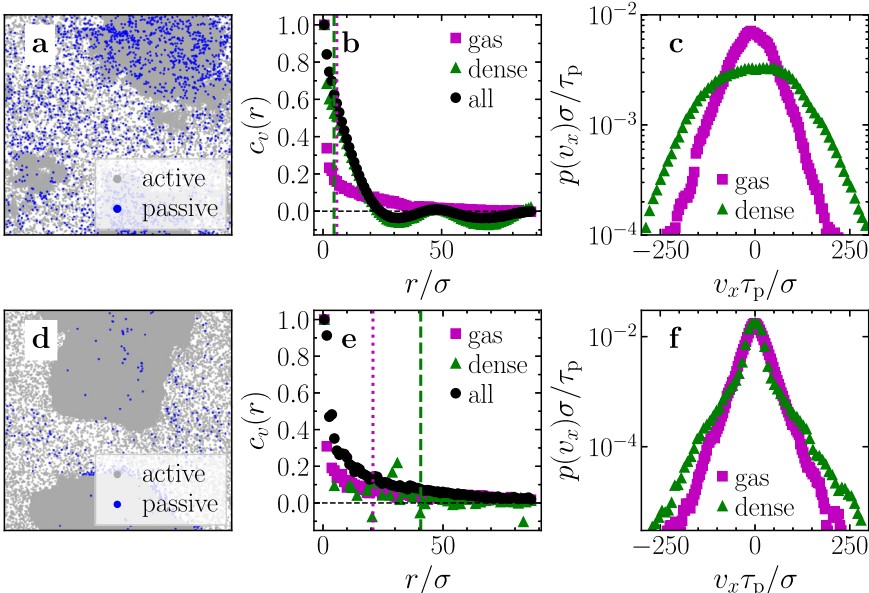

**Fig. 8 | Spatial velocity correlations. a–c** Snapshot in the steady state, spatial velocity correlation function of passive particles (Eq. (6)), and velocity distribution of passive particles in the dense and the gas phase, respectively, for a simulation of N = 20 000 particles at $x_a = 0.90$. The dashed and dotted lines in (**b**) denote the mean distance $\bar{d}$ between passive particles obtained from a Voronoi tessellation in the dense phase ($\bar{d} \approx 4.6\sigma$) and in the gas phase ($\bar{d} \approx 5.8\sigma$), respectively. **d–f** Same as (**a–c**) but for a simulation of N = 100 000 particles at $x_a = 0.996$. The mean distance between the passive particles denoted by the dashed and dotted line in (**e**) is $\bar{d} \approx 41\sigma$ (liquid) and $\bar{d} \approx 21\sigma$ (gas), respectively. All data has been averaged over time in the steady state. Simulation parameters: Pe = 400 (other parameters as in Fig. 2).

within the dense phase. Concretely, we did a simulation with $10^5$ particles and $x_a = 0.996$ at Pe = 400, which again shows MIPS and a significant temperature difference between the passive particles in the dense droplets and the surrounding gas (Fig. 8d–f and Table S1, Supplementary Information). In contrast to the previous scenario shown in Fig. 8a–c, the correlations between passive particles are now significantly reduced, and the mean distance between passive particles in the dense phase is ~41$\sigma$, i.e., larger than the length scale of the velocity correlations (Fig. 8e). Hence, in this parameter regime, the temperature calculation is not much influenced by local collective motion of the passive particles, but remarkably, the passive particles in the dense phase still have a higher temperature than the passive particles in the gas phase. This is shown in Fig. 8f and in Table S1 (Supplementary Information). These results show that the coexistence of hot liquid-like droplets with a colder gas is really induced by the interactions between the active and the passive particles in the dense phase and should also occur for other (fluctuation-based) temperature definitions that are not sensitive to collective motions of the passive particles.

To explicitly see this, we additionally calculated a relative kinetic temperature by using the relative velocity of each particle to the mean velocity of particles in the vicinity

$$k_B T_{\text{kin,rel}} = \frac{m}{2} \left\langle \left(\mathbf{v} - \langle \mathbf{v} \rangle_R \right)^2 \right\rangle, \tag{7}$$

where $\langle \mathbf{v} \rangle_R$ denotes the mean velocity of all particles in a circle of radius $R = 5\sigma$ around the tagged particle. As shown exemplarily in Table S1 (Supplementary Information), the temperature difference is also visible for the relative kinetic temperature, and thus, it is not only a consequence of the observed collective motion but rather a pure effect of the particle interactions. As a result, the key phenomenon of the present work—the coexistence of hot liquid-like droplets and a cold gas—is robust with respect to the choice of the definition of the particle temperature. For a discussion regarding the role of the solvent, we refer the reader to ref. 16.

## Discussion

Mixing overdamped active Brownian particles and underdamped passive Brownian particles leads to a persistent kinetic temperature difference between the dense and the dilute phase when the system undergoes motility-induced phase separation. This temperature difference emerges despite the fact that each of the two components on their own would show a uniform temperature profile. Counterintuitively, the dilute gas-like phase is not always hotter than the dense liquid-like phase but at large Péclet number and fraction of active particles, hot liquid-like droplets can coexist with a cold gas. This temperature reversal results from the competition of two effects: the trapping of passive particles in the dense cluster provokes a cold liquid, whereas the emergence of persistent correlated active–passive particle trajectories in the dense phase primarily heats up the liquid. While the latter effect has not been known in the literature so far, we have shown that it can even overcome the previously discussed trapping effect and lead to the coexistence of a cold gas and hot liquid-like droplets. This phenomenon is robust with respect to the choice of definition of particle temperature and particle-size effects at least up to a size ratio of 10. For even larger size ratios, it can happen that all inertial passive particles remain in the gas phase, and hence, no temperature difference can be observed. Besides their conceptual relevance, our results open a route to create a persistent temperature profile in systems like dusty plasmas or passive granulates by inserting overdamped active particles like bacteria, algae, or synthetic colloidal microswimmers.

## Methods

### Simulation details

The interaction between the particles is modeled by the Weeks–Chandler–Anderson (WCA) potential[59]

$$u(r_{nl}) = \begin{cases} 4\epsilon \left[ \left(\frac{\sigma}{r_{nl}}\right)^{12} - \left(\frac{\sigma}{r_{nl}}\right)^{6} \right] + \epsilon, & r_{nl}/\sigma \leq 2^{1/6} \\ 0, & \text{else} \end{cases}$$

with particle diameter $\sigma$, strength $\epsilon$, and $r_{nl} = |\mathbf{r}_n - \mathbf{r}_l|$. For the simulations with active and passive particles of different diameters, the effective diameter for the interaction between the active and passive particles is chosen as $\sigma_{ap} = (\sigma_a + \sigma_p)/2$, where $\sigma_a$ and $\sigma_p$ denote the diameters of the active and passive particles, respectively. In all simulations, we fix $m_a/(\gamma_t \tau_p) = 5 \times 10^{-5}$, $I/(\gamma_r \tau_p) = 5 \times 10^{-6}$ to recover overdamped dynamics for the active particles[11]. For the passive particles, we fix $m_p/(\tilde{\gamma}_t \tau_p) = 5 \times 10^{-2}$ with the persistence time $\tau_p = 1/D_r$. Furthermore, we set $\epsilon = 10 k_B T_b$, $\tilde{\gamma}_t = \gamma_t$, and $\sigma_a = \sigma_p = \sigma = \sqrt{D_t/D_r}$ (unless otherwise indicated), and we use systems with $N = N_a + N_p$ particles. We choose $\gamma_t = \gamma_r/\sigma^2$ and vary Pe and the fraction $x_a = N_a/(N_a + N_p)$ of the active particles. The total area fraction $\varphi_{tot} = (N_a + N_p) \pi \sigma^2/(4A)$ is set to $\varphi_{tot} = 0.5$, where $A$ denotes the area of the simulation box. The Langevin equations (Eqs. (1)–(3)) are solved numerically in a quadratic box with periodic boundary conditions and with a time step $\Delta t = 10^{-6} \tau_p$ using LAMMPS[60] first for a time of $100\tau_p$ to reach a steady state and afterward for a time of $900\tau_p$ for computing time averages of observables in the steady state.

### Kinetic temperature calculation for dense and dilute phases

To calculate the kinetic temperature of the dense and the dilute phase separately, we distinguish between passive particles in the dense and the gas phase by identifying the largest cluster in the system using the criterion that two particles belong to the same cluster if their distance to each other is smaller than the cutoff distance $r_c = 2^{1/6}\sigma$ of the WCA potential. Then, all particles in the largest cluster are considered as the dense phase, and all other particles as the gas phase (Fig. S7, Supplementary Information). Finally, the kinetic temperature of the passive particles in the dense and the gas phase is obtained by averaging over all passive particles in the dense phase and all passive particles in the gas phase, respectively.

## Data availability

The data that support the findings of this study are available at https://doi.org/10.48328/tudatalib-1389.

## Code availability

The computer codes used for simulations and data analysis are available at https://doi.org/10.48328/tudatalib-1389.

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

## Acknowledgements
The authors gratefully acknowledge the computing time provided to them at the NHR Center NHR4CES at TU Darmstadt (project number p0020259). This is funded by the Federal Ministry of Education and Research, and the state governments participating on the basis of the resolutions of the GWK for national high performance computing at universities (www.nhr-verein.de/unsere-partner). L.H. gratefully acknowledges the support by the German Academic Scholarship Foundation (Studienstiftung des deutschen Volkes).

## Author contributions
B.L. designed the research. L.H. and I.D. performed the research and analyzed the data. L.H. and B.L. discussed the results and wrote the manuscript.

## Funding

## Competing interests
The authors declare no competing interests.
