## [Peer Review File · Nature Communications]

Reviewers' Comments:

Reviewer #1:

Remarks to the Author:

Review report of manuscript titled "Motility-induced coexistence of a hot liquid and a cold gas"

In this manuscript authors have studied the motility induced coexistence of hot liquid and cold gas state in the mixture of active and passive Brownian particles. The state diagram is explored in the phase space of activity and fraction of active particles. The manuscript explored the mechanics of temperature difference for different range of parameter regime. I find the work interesting and content of the manuscript written well. I have following comments before the manuscript is suitable for publication in Journal.

1. The first line of the abstract starts with "Coexisting phase show ...", although rest of the abstract is properly written, but I feel starting the abstract with the complex word like coexistence which itself need some explanation before stating, is not a good idea. Hence I suggest to rewrite the initial part of the abstract.

2. The manuscript discusses the concept of kinetic temperature at various places. It is better to explain it proper context when it is introduced first time.

3. Last paragraph of the introduction should be revised, not clearly written.

4. Detail explanation of kinetic energy difference, is required. The results are not very non-trivial, but they are very robust. Hence some more fundamental understanding is required.

5. In the section mechanism I, "... collisions that occur within ..." it is very clear from the figure 1 (j,k), because temperature is always discussed about the passive particle and here explanation is done using active particle. Hence sentence should be revised.

Also somewhere in the start of the manuscript, it should be made clear that the most of the observations are performed for the tracer or passive particles, if I have understood the manuscript correctly. If I am wrong, then this should be clearly discussed which particles are observed particles.

6. I liked the explanation given in Mechanism -II section. It is very clear and coherently written.

7. Clear mathematical definition of ψ_{loc} is required. I don't find it in the main text as well as in SM.

8. To check the role of size ratio of passive vs. Active particle, two size ratios are considered 1 and 2. But interesting things start for large size ratios. Kindly see the work [Soft Matter, 2018, 14, 6137-6145], hence I feel some comments should be made for more asymmetric size ratios. I know simulations have limitations of using large size ratios, specially when λ_a is large. Still some comments is required.

Also I feel for more asymmetric size ratios the story can be reversed. Hence some comments and also should be discussed in the discussions section.

9. On page 19 .. statement about the new simulations (test one) "To see if these ..." is not very clear, it should be clearly written and its results should be discussed.

Once the above questions and comments are considered, I would like to see the revised manuscript.

Reviewer #2:

Remarks to the Author:

In the present article entitled "Motility-induced coexistence of a hot liquid and a cold gas", the authors report on a novel effect observed in mixtures of overdamped active and underdamped passive particles, namely the coexistence of a hot liquid with a cold gas phase.

The results of the article are very exciting, and give the previously made observation of motility-induced temperature differences in active systems a much broader relevance (including potentially in technical applications) by showing that they occur also in more common particle classes, namely overdamped active and underdamped passive particles (when they are put together). I recommend the article for publication in Nature Communications. Before publication, I suggest the following changes:

- From the abstract, it wasn't immediately clear to me that the article is about overdamped active - underdamped passive mixtures, since the abstract spends a considerable amount of time talking about inertial active matter and never mentions mixtures explicitly.
- I find the structure of the manuscript not ideal. It gives only a very brief indication of the relevant mechanism in the introduction, then (top of p.6) explains a possible mechanism in a lot of detail of which it is then said that it is "only half the story" (does that mean it is wrong?) and then yet later presents the full story. A scientific paper shouldn't have cliffhangers. Present a discussion of the entire relevant mechanism in one subsection of the results section that is easy to find.
- The key governing equations (Eqs. (5) and (6)) are only presented relatively late (in the methods section), which (although these are of course methods) I do not find helpful for a reader who wants to read from the beginning to the end. They should appear in the results section when they are first used.
- Much more importantly, although the central result of the article is that motility-induced temperature differences are possible also in overdamped systems, Eqs. (5) and (6) describe underdamped ABPs (with inertia). Why that? Is it possible to obtain these results also when performing these simulations with zero inertia in the equations of motion? (If not, it would make the results less interesting.)
- In the first section of the results, it is said that the results also hold for particles with very different sizes. Is that supposed to mean that one could also induce such coexistence in, say, water (if one has bacteria swimming in the water)? Or are there limitations making this not possible when the considered particles are water molecules and active colloids - and if so, what are these limitations?

Minor points:

- The initial discussions concerning that "a gas is often hotter than a liquid" are a bit too general- air on a winter day (cold) is certainly cooler than liquid metal (liquid), the point is that usually a gas is hotter than a liquid of the same material.
- The manuscript makes a lot of use of the word "generic" in contexts where I find it at least misleading - for example, it uses "generic active particles" as a synonym for "overdamped active particles", it speaks about "generic effects" (I don't know what that's supposed to mean here), or about "generic components" (does that mean it can be anything)?
- The formulation "transcends a knowledge boundary in the literature" is a bit too poetic in my opinion.
- p. 7 "the majority is" -> "are" and "phases-separated" -> "phase-separated"

- The abstract speaks about examples for inertial active particles giving, e.g., beetles at interfaces as an example, and then mentions that these can show hot-cold-coexistence. Is that supposed to mean that this phenomenon can be observed in beetles?

- SI: "The different persisting temperatures are accompanied by a violation of the equipartition theorem, which holds for systems in equilibrium." -> "The different persisting temperatures are accompanied by a violation of the equipartition theorem, which holds for classical systems in equilibrium." (It doesn't generally hold for quantum systems.)

Response to Reviewer #1:

In this manuscript authors have studied the motility induced coexistence of hot liquid and cold gas state in the mixture of active and passive Brownian particles. The state diagram is explored in the phase space of activity and fraction of active particles. The manuscript explored the mechanics of temperature difference for different range of parameter regime. I find the work interesting and content of the manuscript written well. I have following comments before the manuscript is suitable for publication in Journal.

We thank Reviewer #1 for carefully reading our manuscript and evaluating our work. We are pleased that Reviewer #1 finds our work interesting and suitable for publication in Nature Communications after addressing the Reviewer's specific comments. We thank Reviewer #1 for his/her specific feedback and valuable criticism, and in the following, we would like to address each of the Reviewer's point individually.

The first line of the abstract starts with with "Coexisting phases show ...", although the rest of the abstract is properly written, but I feel starting the abstract with the complex word like coexistence which itself needs some explanation before stating, is not a good idea. Hence I suggest to rewrite the initial part of the abstract.

We agree with the Reviewer and have adapted the first sentence of the abstract as follows:

~~"Coexisting phases show the same~~ If two phases exist at the same time, such as a gas and a liquid, they have the same temperature. This fundamental law of equilibrium physics is known to apply..."

The manuscript discusses the concept of kinetic temperature at various places. It is better to explain it proper context when it is introduced first time.

We thank Reviewer #1 for raising this point. We agree that it might help the reader and improve the text flow if we define and briefly discuss the kinetic temperature when we introduce it for the first time. Accordingly, we have revised the relevant parts of the manuscript. In particular, we have added the following note to the revised introduction and have now defined kinetic temperature at the first occurrence in the results section of the main text.

"... Despite this, very recently, it was found that at phase coexistence in certain active systems consisting of active particles which consume energy from their environment to propel themselves [12–14], the dilute (gas-like) phase is hotter by up to one or two orders of magnitude compared to the dense (liquid-like) phase in terms of the kinetic temperature ~~of the particles [11,15,16].~~ [11,15,16]. (Here, the kinetic temperature is defined as the mean kinetic energy per particle, which is equivalent to other temperature definitions in equilibrium.) While this complies with our intuition that gases..."

“...As for systems of active overdamped particles alone [11], we find that the active and passive particles in both phases have the same kinetic temperature (shown in Fig. 1d for the passive particles). Here, following Refs. [11,16,63,64], we define the temperature of the particles based on their kinetic energy as $k_B T_{\text{kin}}^{a/p} = m_{a/p} \langle (\vec{v} - \langle \vec{v} \rangle)^2 \rangle / 2$, which is well-defined also in non-equilibrium systems [65]. (Note that the phenomena which we report occur similarly if using other temperature definitions such as the Maxwell-Boltzmann temperature, as further discussed below.) Let us now explore if...”

Last paragraph of the introduction should be revised, not clearly written.

We thank Reviewer #1 for this suggestion and revised the last paragraph of the introduction. All changes are marked within the red-line version of our revised manuscript (we do not show it here because it is a rather long paragraph).

Detailed explanation of kinetic energy difference, is required. The results are not very non-trivial, but they are very robust. Hence some more fundamental understanding is required.

Following the Reviewer’s comment below regarding the Mechanism-II section, in which we explain the hot-liquid–cold-gas coexistence in detail, we assume that Reviewer #1 is fully satisfied with the explanation of the mechanism underlying the coexistence of a hot liquid and a cold gas. Accordingly, here, we only refer to the mechanism underlying the coexistence of a cold liquid and a hot gas.

Indeed, the emergence of a kinetic temperature difference is very robust. We of course agree with the Reviewer that a fundamental understanding of the mechanism leading to the cold-liquid–hot-gas coexistence is important, and accordingly, we explore and discuss it in detail in our manuscript. Following the Reviewer’s remarks, we have made an effort to go even deeper and have performed additional simulations to verify an aspect that we have not explicitly explored before.

First, for weak or moderate self-propulsion of the active particles, the dense phase is colder than the dilute one in terms of the kinetic temperature of the passive particles. This is because the passive particles are essentially trapped within the dense phase, and the multiple collisions with surrounding particles make them change their direction of motion frequently such that they have no space to speed up [see also S. Mandal, B. Liebchen, and H. Löwen, Phys. Rev. Lett. 123, 228001 (2019)]. In contrast, in the gas phase, an active particle can push a passive one persistently forward when colliding with a passive particle. This process speeds up the passive particles in the gas phase and leads to the coexistence of a hot gas and a cold liquid. We note that a key ingredient of this mechanism, which we had not explicitly resolved before, is that passive particles remain trapped and densely packed within the dense phase for a long time. However, for inertial passive particles, it is not immediately clear that this is the case. To explicitly resolve this, we have performed additional simulations to determine the effective force on the passive particles.

These simulations are performed in a rectangular simulation box to ensure that the border of the clusters are approximately stationary (which allows to average over time in

the steady state for obtaining a force profile at the border of the dense phase). We have calculated the total interaction force on each passive particle and coarse-grained the resulting forces using Gaussian-kernel density estimation. The time averaged force field is shown in Fig. R1: While the (averaged) effective force is approximately zero within the dense and the dilute phase, a strong effective force that points to the center of the cluster emerges on the cluster boundary. This force pushes the passive particles together within the dense phase and “traps” them. Without such inwards-pointing force on the passive particles, one could end up with a situation where (almost) no passive particles are in the dense phase (which can be the case for very large size ratios between the two species, see Fig. R2), and hence, there would no longer be a coexistence of two different temperatures.

To clarify the underlying mechanism for the reader, we slightly revised the section “Coexistence of a hot gas and a cold liquid” to make it clearer to a broad readership. All changes are marked in the red-line version of our revised manuscript. Additionally, to make the results for the effective forces also available to the readers of our manuscript, we have added the following discussion to the Supplemental Material together with Fig. R1 (which we have added as Fig. S12 in the Supplemental Material):

“At low and intermediate Pe , we have shown that the passive particles are colder in the dense phase compared to the dilute phase in terms of their kinetic temperature. As described in the main text, the mechanism of this phenomenon is based on a trapping of inertial passive particles within the dense phase. In contrast, in the dilute phase, active particles can push passive particles forward and persistently speed them up. One key ingredient of this mechanism is that passive particles remain trapped and are densely packed within the dense phase, which is not trivial to be the case for inertial passive particles. To explore this in more detail, we now calculate the effective force acting on the passive particles depending on their position. To this end we made a simulation in a slit geometry in which the border of the dense phase is approximately stationary and does not move much, which allows to perform a long-time average. As shown in Fig. S12b, the effective force always points to the dense phase at its border, i.e., the passive particles are pushed inside the dense phase. This effective force finally ensures, that the passive particles remain densely packed within the dense phase.”

Fig. R1: **Effective force on passive particles.** **a** Snapshot of the binary mixture in a thin rectangular box showing motility-induced phase separation. The box shape ensures that the border of the dense phase is approximately stationary allowing for long-time averages. Here, we show an extract from the simulation showing one interface and a part of the dense and the dilute phase in its vicinity. **b** Corresponding coarse-grained force field of the interaction force from the Weeks-Chandler-Anderson (WCA) potential acting on the passive particles. The color and arrow length represent the strength of the effective force, the orientation of the white arrows its direction. A strong effective force is pushing passive particles towards the dense phase (yellow region). Parameters: $Pe = 100$, $x_a = 0.6$, $L_x = 784\sigma$, $L_y = 40\sigma$ (other parameters as in Fig. 2 in the main text).

Fig. R2: **Steady-state snapshot at large size ratio.** Parameters: $Pe = 100$, $x_a = 0.99$, $s = \sigma_p/\sigma_a = 10$, $\phi_{tot} = 0.7$.

In the section mechanism I, “.. collisions that occur within” it is very clear from the figure 1 (j,k), because temperature is always discussed about the passive particle and here explanation is done using active particle. Hence sentence should be revised.

We thank Reviewer #1 for noticing this point. We agree that the wording is a bit misleading. Indeed, the discussion about temperature differences in pure active systems is not required at this point. Therefore, we deleted the first part of this section:

~~“In single component systems made of inertial active Brownian particles, it has been shown that the frequent collisions that occur within the dense phase effectively slow down the active particles [11,16]. Effectively, this is similar to the effect of inelastic collisions in driven granular particles [39] and fully consistent with our observations at moderate Péclet numbers. We now explore the mechanism underlying...”~~

I liked the explanation given in Mechanism -II section. It is very clear and coherently written.

We thank Reviewer #1 for this positive feedback.

Clear mathematical definition of ϕ_{loc} is required. I don't find it in the main text as well as in SM.

We thank the Reviewer for noticing that the mathematical definition of the local area fraction is indeed missing. We have added it to the main text as follows:

~~“... The transition line between the uniform and the MIPS regime is obtained by analyzing the distribution of the local area fraction ϕ_{loc} , which is $\phi_{\text{loc}}(\vec{r}_i) = \sum_{j=1}^N \sigma_j^2 H(R - |\vec{r}_i - \vec{r}_j|) / (4R^2)$, where H is the Heaviside step function and σ_j the diameter of particle j , calculated from averages over circles of radius $R = 5\sigma$. Its distribution is unimodal in the uniform regime and bimodal in the coexistence regime ...”~~

To check the role of size ratio of passive vs. Active particle, two size ratios are considered 1 and 2. But interesting things start for large size ratios. Kindly see the work [Soft Matter, 2018, 14, 6137-6145], hence I feel some comments should be made for more asymmetric size ratios. I know simulations have limitations of using large size ratios, specially when x_a is large. Still some comments is required. Also I feel for more asymmetric size ratios the story can be reversed. Hence some comments and also should be discussed in the discussions section.

We thank Reviewer #1 for raising this interesting point. As known from the mentioned reference [Soft Matter, 2018, 14, 6137-6145] (cited in the manuscript as Ref. [72]), different size ratios can lead to different effects within mixtures of active and passive Brownian

particles such as the tuning from uniform to clustered and even phase-separated states and effective attractions between passive particles induced by the interactions with surrounding active particles. In the previous version of our manuscript, we have shown that our results are robust against the variation of the size ratio up to $s = \sigma_p/\sigma_a = 4$. Since the temperature difference is mainly caused by processes in which active particles push passive particles forward, we do not expect that the observation of a kinetic temperature difference breaks down for larger size ratios. To show this explicitly, we have done new simulations with size ratios up to $s = \sigma_p/\sigma_a = 10$ and up to $N = 10^6$ particles as well as for a large fraction of active particles of $x_a = 0.99$. The results are summarized in Fig. R3. Most importantly, the observation of kinetic temperature differences is robust against the size ratio even for large size ratios of up to $s = 10$.

Similarly, also the opposite case of a hot liquid that coexists with a colder gas is robust and leads to a temperature difference of $T_{\text{kin}}^{(\text{dense})} - T_{\text{kin}}^{(\text{gas})} \approx 196.0T_b$ for $s = 10$ for example.

In addition, it is interesting to note that there is also the following side effect occurring for large size differences: Large passive particles tend to accumulate in the gas phase and do not stay within the dense phase at large Péclet numbers of the active particles, i.e., for very fast active particles. This effect might ultimately prevent the observation of a kinetic temperature difference at very large $s \gg 10$ and large Pe because there won't be any passive particles within the dense phase. Interestingly, this is in contrast to purely overdamped mixtures, where large size ratios support the formation of large clusters of passive particles [Soft Matter, 2018, 14, 617-6145; Phys. Rev. E 108, 034603 (2023)].

We have updated Fig. 7 of the main text with Fig. R3 and have added the blue marked text in the following paragraphs within the section ‘‘Role of the particle size’’ and the discussion section, respectively:

‘‘For simplicity, we have considered active and passive particles with the same size and the same drag coefficients so far but with significantly different material density. Now, we show that persistent temperature differences also occur when the passive particles are significantly larger than the active ones. We have varied the ratio σ_p/σ_a size ratio $s = \sigma_p/\sigma_a \in \{1, 2, 3, 4, 6, 8, 10\}$ keeping σ_a as well as m_p and m_a fixed. For the γ and ϕ_{tot} fixed. The fraction x_a is chosen such that the area fraction of active particles is approximately 0.5 for small size ratios. For large size ratios, we kept $x_a = 0.99$ fixed to ensure that enough passive particles are inside the system. For the drag coefficient of the passive particles, we choose $\tilde{\gamma}_t = \sigma_p \gamma_t / \sigma_a$. Our results are exemplarily shown in Fig. 7 for $\text{Pe} = 100$. Here, we observe a persistent kinetic temperature difference between the passive particles in the dense and the gas phase even for significantly different particle sizes (Fig. 7h). This temperature difference is also visible in the velocity distributions, which are broader in the gas phase compared to the dense phase (Fig. 7e,f). Hence, the observation of a cold dense phase that coexists with a hotter surrounding gas persists even for large size ratios. The opposite case, i.e., hot liquid-like droplets coexisting with a colder gas, is also robust and leads to a temperature difference of $T_{\text{kin}}^{(\text{dense})} - T_{\text{kin}}^{(\text{gas})} \approx 196.0T_b$ for $\sigma_p/\sigma_a = 10$ at $\text{Pe} = 400$, $x_a = 0.99$, and $\phi_{\text{tot}} = 0.70$ for example. Notice however, that very large passive particles tend to accumulate in the dilute phase especially at large Pe making it challenging to calculate a precise value of the temperature difference. Interestingly, this is in contrast to purely overdamped mixtures, where large size ratios support the formation of large passive-particle clusters [49,72].’’

“... This phenomenon is robust with respect to the choice of definition of particle temperature and particle-size effects at least up to a size ratio of 10. For even larger size ratios, it can happen that all inertial passive particles remain in the gas phase, and hence, no temperature difference can be observed. Besides their conceptual relevance, our results open a route ...”

Fig. R3: **Effect of the particle size.** **a–d** Simulation snapshots with four different sizes of the passive particles as given in the panel titles. The passive particles are colored with respect to their kinetic temperature. **e,f** Velocity distribution of the passive particles in the dense and the gas phase, respectively, for different size ratios as indicated in the key. **g** Mean-square displacement of the active (dashed lines) and passive (solid lines) particles. **h** Corresponding kinetic temperatures of passive particles in the dense and the gas phase. Parameters: $Pe = 100$, $\tilde{\gamma}_t = \sigma_p \gamma_t / \sigma_a$, $\phi_{tot} = 0.7$, $N_a + N_p = 2 \times 10^4$ for $\sigma_p/\sigma_a = 1, 2, 3, 4$ and $N_a + N_p = 5 \times 10^4$ for $\sigma_p/\sigma_a = 6, 8, 10$ (other parameters as in Fig. 2).

On page 19 .. statement about the new simulations (test one) “To see if these ...” is not very clear, it should be clearly written and its results should be discussed.

Following the Reviewer’s suggestion, we have revised the relevant paragraph and have added additional detail to the discussion. The revised text is as follows:

“... As shown in Fig. 8b (and Movie S4, Supplemental Material), velocity correlations are indeed present between the passive particles in the dense phase over a significant spatial range –The indicating that collective motion might strongly influence the kinetic temperatures. In fact, we find that the mean distance between the passive particles in the dense phase calculated from a Voronoi tessellation is given by approximately 4.6σ for the case shown in Fig. 8a–c and therefore, much smaller than the length scale of the velocity correlations –To see if these spatial correlations are crucial for the emergence of a

~~temperature difference, we~~ (Fig. 8b). To exclude that such collective motions are required to achieve a coexistence of a hot liquid and a cold gas, we have performed simulations with a very low fraction of passive particles such that their typical distances to each other are significantly longer than the velocity correlations within the dense phase. Concretely, we did a simulation with 10^5 particles and $x_a = 0.996$ at $Pe = 400$, which again shows MIPS and a significant temperature difference between the passive particles in the dense droplets and the surrounding gas (Fig. 8d–f). ~~Here and Tab. S1, Supplemental Material).~~ In contrast to the previous scenario shown in Fig. 8a–c, the correlations between ~~PBPs are significantly reduced~~ passive particles are now significantly reduced, and the mean distance between passive particles in the dense phase is approximately 41σ . ~~In, i.e.,~~ larger than the length scale of the velocity correlations (Fig. 8e). Hence, in this parameter regime, the temperature calculation is not much influenced by local collective motion of the ~~PBPs. Remarkably, as demonstrated in Fig. 8f and as shown in Tab. S1 (Supplemental Material)~~ passive particles, but remarkably, the passive particles in the dense phase still have a higher temperature than the passive particles in the gas phase. This ~~shows is shown~~ in Fig. 8f and in Tab. S1 (Supplemental Material). These results show that the coexistence of hot liquid-like droplets ...”

Response to Reviewer #2:

In the present article entitled “Motility-induced coexistence of a hot liquid and a cold gas”, the authors report on a novel effect observed in mixtures of overdamped active and underdamped passive particles, namely the coexistence of a hot liquid with a cold gas phase. The results of the article are very exciting, and give the previously made observation of motility-induced temperature differences in active systems a much broader relevance (including potentially in technical applications) by showing that they occur also in more common particle classes, namely overdamped active and underdamped passive particles (when they are put together). I recommend the article for publication in Nature Communications. Before publication, I suggest the following changes:

We thank Reviewer #2 for carefully evaluating our work and highlighting the significance and importance of our results for a broad readership and potential technical applications. We also thank Reviewer #2 for his/her very useful suggestions, which helped us to further improve the manuscript.

From the abstract, it wasn't immediately clear to me that the article is about overdamped active - underdamped passive mixtures, since the abstract spends a considerable amount of time talking about inertial active matter and never mentions mixtures explicitly.

We thank Reviewer #2 for pointing this out and have modified the abstract as follows:

“... Here, we show that a kinetic temperature difference across coexisting phases can occur even in equilibrium systems when adding generic (overdamped) self-propelled particles. In ~~this case, surprisingly, particular, we consider mixtures of overdamped active and inertial passive Brownian particles and show that when they phase separate into a dense and a dilute phase, both phases have different kinetic temperatures. Surprisingly,~~ we find that the dense phase (liquid) cannot only be colder but also hotter than the dilute phase (gas). This ~~generic~~ effect hinges on ...”

I find the structure of the manuscript not ideal. It gives only a very brief indication of the relevant mechanism in the introduction, then (top of p.6) explains a possible mechanism in a lot of detail of which it is then said that it is "only half the story" (does that mean it is wrong?) and then yet later presents the full story. A scientific paper shouldn't have cliffhangers. Present a discussion of the entire relevant mechanism in one subsection of the results section that is easy to find.

We very much thank the Reviewer for this valuable feedback and agree that the story would be clearer for the reader if we rearrange some parts of the manuscript. To address the Reviewer's concern, we have combined the two sections “Mechanism I” and “Mechanism II” to one section that now contains the complete explanation of the mechanism

that causes the coexistence of hot liquid-like droplets and a colder surrounding gas. Additionally, we now explain the mechanism for the hot-gas cold-liquid coexistence within the section “Coexistence of a hot gas and a cold liquid”. Due to the length of the modified text, we do not provide a copy of all changes here but all changes are marked in the red-line version.

The key governing equations (Eqs. (5) and (6)) are only presented relatively late (in the methods section), which (although these are of course methods) I do not find helpful for a reader who wants to read from the beginning to the end. They should appear in the results section when they are first used.

We thank Reviewer #2 for this suggestion and moved Eqs. (5) and (6) to the model section in the main part of the manuscript, where we introduce the model for the first time. The Methods section now only focuses on simulation details rather than the description of the whole model. Again, due to the length of the model section, we do not provide a copy of all changes here but they are marked in the red-line version of our revised manuscript.

Much more importantly, although the central result of the article is that motility-induced temperature differences are possible also in overdamped systems, Eqs. (5) and (6) describe underdamped ABPs (with inertia). Why that? Is it possible to obtain these results also when performing these simulations with zero inertia in the equations of motion? (If not, it would make the results less interesting.)

We thank Reviewer #2 for raising these two important questions. First, we are working in the strongly overdamped regime rather than in the overdamped limit because we would like to not only have access to the velocities of the passive particles but also of the active particles to be able to calculate their kinetic temperature and velocity distributions as well. In simulations with zero inertia (i.e., using the overdamped Langevin equations), velocity is not well defined. Therefore, we use the underdamped Langevin equations with a very small mass to obtain overdamped dynamics with access to the exact velocities of all particles. To answer the second question and to show that our results can be reproduced even when considering zero inertia in the Langevin equation of the active particles, we have done additional simulations for the case of a hot gas coexisting with a cold liquid as well as for the opposite case of a cold gas coexisting with a hot liquid. The results are shown in Fig. R4 in form of a coarse-grained kinetic temperature field of the passive particles, for which we still have access to the exact velocities (since the passive particles are inertial). The parameters are all the same as in Fig. 1 of the main text except that $m_a = 0$ and $I = 0$. Similar to the effectively overdamped simulations, a kinetic temperature difference can be observed between passive particles in the dense and the dilute phase in both parameter regimes. Thus, we obtain the same results when performing the simulations with zero inertia.

To make these additional results also available to the readers of our manuscript, we added Fig. R4 to the Supplemental Material and added a short note in the main text:

“... and k_B is the Boltzmann constant. The corresponding Langevin equations see Methods Eqs To have access to a well-defined instantaneous particle velocity, we explicitly account for inertia for the active species but choose a very small mass to stay in the overdamped regime. Notice that using overdamped Langevin equations instead (\$m_a = 0\$ ) essentially yields the same results (Fig. (S13, Supplemental Material).
The passive particles ...”

Fig. R4: **Zero-inertia limit for active particles.** To show that our results persist even when considering zero inertia for the active particles, we also made simulations using the overdamped Langevin equation for the active particles for the scenario **a** cold-liquid–hot-gas and the opposite scenario **b** hot-liquid–cold-gas. Here, we show the corresponding coarse-grained kinetic temperature fields. The white lines denote the border of the dense phase, which is located inside the area enclosed by the white lines. Parameters used in panels a and b are the same as in Fig. 1e–h and i–l in the main text, respectively, but with $m_a/(\gamma\tau_p) = 0$ and $I/(\gamma\tau_p) = 0$.

In the first section of the results, it is said that the results also hold for particles with very different sizes. Is that supposed to mean that one could also induce such coexistence in, say, water (if one has bacteria swimming in the water)? Or are there limitations making this not possible when the considered particles are water molecules and active colloids - and if so, what are these limitations?

We thank the Reviewer for raising these questions and bringing up these thoughts. As we show in the section “Role of inertia” as well as in Fig. S3 in the Supplemental Material, the passive particles have to be inertial (i.e., underdamped) to obtain a kinetic temperature difference. Hence, if we consider overdamped active particles such as active colloids, the surrounding passive particles have to be significantly heavier than the active particles to be able to observe different kinetic temperatures in coexisting phases. This can be reached by either choosing a much higher material density for the passive particles or by making them much larger than the active species. These requirements preclude the possibility to induce such temperature coexistence in water for example because the water molecules are much lighter than active colloids.

The initial discussions concerning that "a gas is often hotter than a liquid" are a bit too general- air on a winter day (cold) is certainly cooler than liquid metal (liquid), the point is that usually a gas is hotter than a liquid of the same material.

We thank Reviewer #2 for this useful comment and clarified this point in the initial part of the introduction as follows:

“We are all used to the experience that a gas is often hotter than a liquid of the same material. For example, to evaporate [...] The central exception from the experience that gases are hotter than liquids of the same material occurs when two phases ...”

The manuscript makes a lot of use of the word “generic” in contexts where I find it at least misleading - for example, it uses “generic active particles” as a synonym for “overdamped active particles”, it speaks about “generic effects” (I don’t know what that’s supposed to mean here), or about “generic components” (does that mean it can be anything)?

Following the Reviewer’s comments, we have removed the word “generic” in most places or have replaced it with more specific terms where possible. We have marked all corresponding changes in the revised manuscript.

The formulation “transcends a knowledge boundary in the literature” is a bit too poetic in my opinion.

We thank Reviewer #2 for this feedback and removed this part in the text.

p. 7 “the majority is” -> “are” and “phases-separated” -> “phase-separated”

We thank the Reviewer for noticing these typos and corrected them accordingly.

The abstract speaks about examples for inertial active particles giving, e.g., beetles at interfaces as an example, and then mentions that these can show hot-cold-coexistence. Is that supposed to mean that this phenomenon can be observed in beetles?

The hot-cold coexistence should be generally observable in inertial active particles undergoing phase separation (e.g., through motility-induced phase separation), or at least clustering. Examples for such inertial active particles that can phase separate are active granular particles (also called microflyers), active Janus colloids in a plasma as well as beetles at interfaces. Thus, we expect that the hot-cold coexistence should be generally observable in beetles at interfaces.

Still, following the Reviewer's question and the fact that literature on the latter example is relatively sparse, we have removed this example from the abstract. Instead, we now mention the beetles in the main text together with the following reference related to beetles showing motility-induced phase separation [H. L. Devereux, C. R. Twomey, M. S. Turner, and S. Thutupalli, Whirligig Beetles as Corralled Active Brownian Particles, J. R. Soc. Interface 18, 20210114 (2021).]

“... This fundamental law of equilibrium physics is known to apply even to many non-equilibrium systems. However, recently, there has been much attention in the finding that inertial self-propelled particles like Janus colloids in a plasma ~~, microflyers, or beetles at interfaces or microflyers~~ could self-organize into a hot gas-like phase that coexists with a colder liquid-like phase. Here, we show ...”

“... Accordingly, the temperature difference must ~~somehow~~ arise from the interactions of the two species, ~~as we will explore in more detail below. It is tempting to relate our observation of a temperature difference at the level of the particles.~~ To understand this in detail, it first might be tempting to start from the common understanding of kinetic temperature differences in granular systems or purely active systems made of inertial ABPs such as Janus colloids in a plasma [68], microflyers [69], or beetles at interfaces [70] which relates the emergence of a temperature difference to an enhanced energy dissipation in the dense phase ... ”

SI: “The different persisting temperatures are accompanied by a violation of the equipartition theorem, which holds for systems in equilibrium.” -> “The different persisting temperatures are accompanied by a violation of the equipartition theorem, which holds for classical systems in equilibrium.” (It doesn't generally hold for quantum systems.)

We thank Reviewer #2 for this important remark and changed the text accordingly.

Reviewers' Comments:

Reviewer #1:

Remarks to the Author:

Authors have revised the manuscript as the changes suggested by me. Hence I feel now the manuscript is suitable for the publication in the Journal.

Reviewer #2:

Remarks to the Author:

The authors have addressed all points I have raised in a carefully and convincing way. I congratulate them on their results and recommend the article for publication in "Nature Communications" in the present form.